# Queuosine biosynthetic enzyme, QueE moonlights as a cell division regulator

**Samuel A. Adeleye, Srujana S. Yadavalli** [ID]*

Waksman Institute of Microbiology and Department of Genetics, Rutgers University, Piscataway New Jersey, United States of America

* sam.yadavalli@rutgers.edu

**Data Availability Statement:** The authors declare that all data supporting the findings of this study are available within the article and its Supplementary Information files.

## Abstract

In many organisms, stress responses to adverse environments can trigger secondary functions of certain proteins by altering protein levels, localization, activity, or interaction partners. *Escherichia coli* cells respond to the presence of specific cationic antimicrobial peptides by strongly activating the PhoQ/PhoP two-component signaling system, which regulates genes important for growth under this stress. As part of this pathway, a biosynthetic enzyme called QueE, which catalyzes a step in the formation of queuosine (Q) tRNA modification is upregulated. When cellular QueE levels are high, it co-localizes with the central cell division protein FtsZ at the septal site, blocking division and resulting in filamentous growth. Here we show that QueE affects cell size in a dose-dependent manner. Using alanine scanning mutagenesis of amino acids in the catalytic active site, we pinpoint residues in QueE that contribute distinctly to each of its functions–Q biosynthesis or regulation of cell division, establishing QueE as a moonlighting protein. We further show that QueE orthologs from enterobacteria like *Salmonella typhimurium* and *Klebsiella pneumoniae* also cause filamentation in these organisms, but the more distant counterparts from *Pseudomonas aeruginosa* and *Bacillus subtilis* lack this ability. By comparative analysis of *E. coli* QueE with distant orthologs, we elucidate a unique region in this protein that is responsible for QueE's secondary function as a cell division regulator. A dual-function protein like QueE is an exception to the conventional model of "one gene, one enzyme, one function", which has divergent roles across a range of fundamental cellular processes including RNA modification and translation to cell division and stress response.

## Author summary

In stressful environments, proteins in many organisms can take on extra roles. When *Escherichia coli* bacteria are exposed to antimicrobial compounds, the cell activates the PhoQ/PhoP signaling system, increasing the production of an enzyme called QueE. QueE is usually involved in the formation of queuosine (Q) tRNA modification. When cells make abundant QueE, it interacts with a vital division protein, FtsZ, disrupting division and causing elongation – a "moonlighting" function. Detailed study of QueE reveals specific regions involved in Q biosynthesis or cell division. QueE in organisms closely related to *E. coli* also has dual roles, while distant relatives are unifunctional. Comparative analysis

**Funding:** SSY received the National Institutes of Health - National Institute of General Medical Sciences (NIH-NIGMS) ESI-MIRA R35 GM147566 and institutional start-up funds from Rutgers. SAA was supported by the Waksman Institute Busch Predoctoral fellowship. https://grants.nih.gov/grants/guide/pa-files/PAR-23-145.html The funders did not play any role in the study design, data collection and analysis, decision to publish, or preparation of the manuscript.

**Competing interests:** I have read the journal's policy and the authors of this manuscript have the following competing interests: Srujana S. Yadavalli consults for and collaborates with Designs for Vision Inc. and Samuel A. Adeleye declared that no competing interests exist.

identifies a unique *E. coli* QueE region regulating cell division. This study shows QueE's versatility in linking and impacting distinct cellular processes such as RNA metabolism, protein translation, cell division, and stress response.

## Introduction

Moonlighting proteins are multifunctional molecules that challenge the one-gene-one-function paradigm [1,2]. They are prevalent across all kingdoms of life, from humans to bacteria. In bacteria, they have been shown to link vital metabolic processes to physiological stress responses such as regulation of cell size [3], adhesion [4], bacterial virulence, and pathogenicity [5,6]. Moonlighting proteins associated with cell division are particularly intriguing because they connect this fundamental process to many additional networks that can profoundly influence cellular physiology. Specific examples include glucosyltransferases–OpgH in *Escherichia coli* and UgtP in *Bacillus subtilis*, which modulate cell division based on nutrient availability [3,7,8]. In *E. coli*, DnaA, known for its role in initiating DNA replication, also acts as a transcription factor regulating gene expression [7]. In *B. subtilis*, protein DivIVA has been implicated in chromosome segregation and spore formation apart from its role in cell division [9]. Studying moonlighting proteins provides a fascinating path to understanding protein functionality, interactions, and complexity within a cell. While our understanding of moonlighting proteins in bacteria has advanced significantly in recent years, several knowledge gaps and challenges remain in this field of research. One of the primary knowledge gaps is the precise molecular mechanisms underlying moonlighting. While some moonlighting functions have been identified and characterized, we often lack a comprehensive understanding of how a single protein can perform multiple functions.

Our previous work found that an enzyme–QueE–involved in the biosynthetic pathway for queuosine (Q) tRNA modification also plays a role in stress response [10]. Q is a hypermodified guanosine that is found ubiquitously at the wobble position of the anticodon loop of specific tRNAs–$tRNA^{His}$, $tRNA^{Tyr}$, $tRNA^{Asp}$, and $tRNA^{Asn}$ [11–14]. Q-tRNA modification is crucial to maintain translation fidelity and efficiency [15]. It has also been implicated in redox, virulence, development, and cancers [14,16,17]. Despite its universal distribution and importance, only bacteria are capable of *de novo* synthesis of Q from guanosine triphosphate (GTP), and eukaryotes salvage precursors of Q from diet or gut bacteria [11,14]. In the biosynthesis of Q, three enzymes QueD, QueE, and QueC, are required to produce a vital intermediate $PreQ_0$ [18]. Specifically, QueE (also called 7-carboxy-7-deazaguanine or CDG synthase) catalyzes the conversion of the substrate $CPH_4$ (6-carboxy-5,6,7,8-tetrahydropterin) to CDG (7-carboxy-7-deazaguanine) [19,20]. The role of QueE in the biosynthesis of Q has been well characterized, and crystal structures of QueE homologs from *Bacillus subtilis*, *Burkholderia multivorans*, and *Escherichia coli* have been solved, providing insights into the catalytic mechanism [19–22]. More recently, a second function for QueE has been described during the stress response of *E. coli* cells exposed to sub-MIC levels of cationic-antimicrobial peptides (AMP) [10]. During this response, the PhoQ/PhoP two-component signaling system, which plays an important role in sensing antimicrobial peptides and several other signals [23–25] is strongly activated, leading to an increase in QueE expression (Fig 1A). When QueE is upregulated in the cell, it binds at the site of cell division and blocks septation. Consequently, *E. coli* cells grow as long heterogeneous filaments ranging from a few microns to hundreds of microns in length. It has been shown that QueE localizes to the septal Z-ring, a vital structure in bacterial cell division [26,27], inhibiting septation post-Z ring formation in an SOS-independent manner [10]. This

QueE-mediated filamentation phenotype is also observed under other conditions that activate the PhoQ/PhoP system robustly, such as when cells lacking MgrB (a negative feedback inhibitor of PhoQ [28,29]) are grown under magnesium limitation. Although historically filamentation was considered a sign of death, it can also be a crucial adaptive response to stress [30–33].

In this study, using QueE as a model, we investigate the molecular determinants that allow a protein to perform two distinct functions. We analyze QueE's dual roles in tRNA modification and cell division by utilizing a specialized northern blotting technique and microscopy, respectively, as readouts. By examining single alanine mutants and variants of *E. coli* QueE (*Ec*QueE), we establish that the catalytic activity required for QueE's biosynthetic role is dispensable for its function as a cell division regulator. As a corollary, we identify several individual amino acid residues and a specific region in *Ec*QueE, which are necessary to cause filamentation but not for Q biosynthesis. Analysis of QueE homologs from different bacteria shows that the moonlighting functions of QueE are conserved among other enterobacteria, suggesting this mode of cell division regulation is widespread.

## Results

### Increased expression of QueE leads to its secondary activity in *E. coli* cell division

To study the molecular determinants in QueE contributing to its dual functions, we utilize two distinct readouts (Fig 1B). Firstly, to test QueE's function in the formation of Q modification, we adapted published methods based on a specialized gel containing N-acryloyl-3-aminophenylboronic acid (APB) and northern blotting to detect queuosinylated tRNAs (Q-tRNAs) [34,35]. In this technique (APB gel and northern blotting), modified Q-tRNAs containing cis-diol reactive groups migrate slower than unmodified guanosinylated-tRNA (G-tRNAs) [35]. Secondly, to examine QueE's ability to cause filamentation, we use phase contrast microscopy and monitor cell morphology. Using Δ*queE* cells harboring *queE* on an IPTG-inducible plasmid, we analyzed Q-tRNA formation and filamentation in the presence or absence of the inducer. Wild-type (WT) cells carrying an empty plasmid were included as a control. As a positive control, WT cells show the formation of Q-tRNA$^{Tyr}$ (Fig 1C). As expected, Δ*queE* cells carrying an empty vector show a band corresponding to unmodified guanosinylated-tRNA$^{Tyr}$ (G-tRNA$^{Tyr}$) but not queuosinylated-tRNA$^{Tyr}$ (Q-tRNA$^{Tyr}$), confirming that QueE is required to produce Q. The complementation of Δ*queE* cells with WT QueE restores the formation of intact Q-tRNA$^{Tyr}$ regardless of induction, indicating that the overexpression of the enzyme is not necessary for the complementation of Q-tRNA synthesis activity. Regarding cell morphology, cells lacking QueE do not filament. Δ*queE* cells expressing basal levels of QueE in the absence of inducer also do not filament, however, Δ*queE* cells induced to express WT QueE show filamentation (Fig 1D,[10]), implying that increased expression of QueE is needed for cell division inhibition.

### QueE regulates cell length in a dose-dependent manner

Expression of *queE* is upregulated during filamentation of *E. coli* cells in response to sub-lethal concentrations of a cationic antimicrobial peptide, C18G [10]. These elongated cells vary from ~2 to hundreds of microns in length, with an average size of ~20 μm. Interestingly, *queE* expression from an IPTG-inducible promoter on a plasmid in wild-type (WT) cells also causes filamentation. To systematically examine the effect of increasing QueE expression on cell length, we performed an IPTG titration using the inducible plasmid encoding *E. coli* QueE (*Ec*QueE) in MG1655 Δ*lacZYA* cells. As the IPTG concentration is increased from 0 to

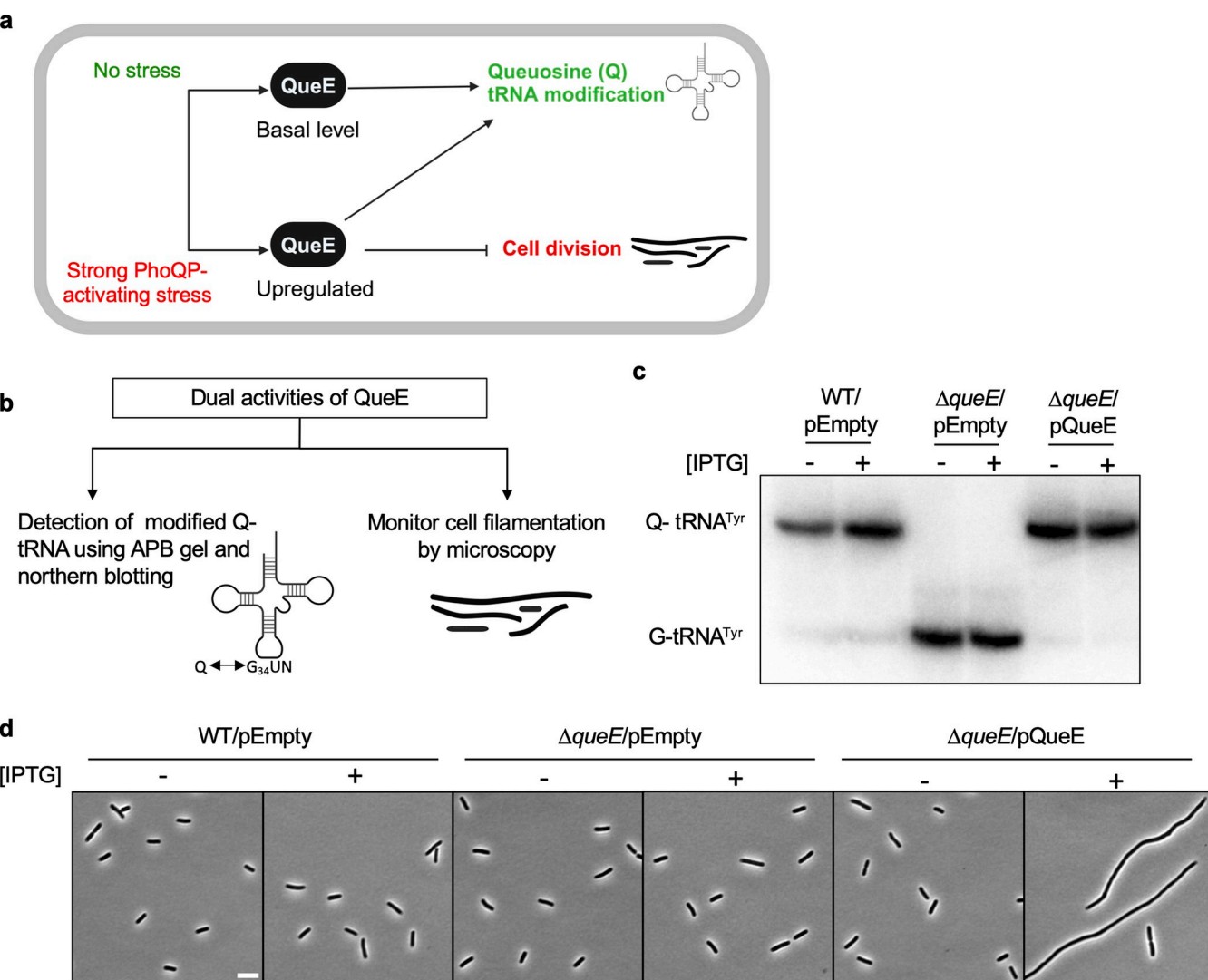

**Fig 1. Dual roles of *E. coli* QueE (*Ec*QueE) during stress response.** (a) Schematic representation of the dual roles of QueE upon strong activation of the PhoQ/PhoP two-component system. QueE is an enzyme required for the biosynthesis of queuosine (Q) tRNA modification. *E. coli* cells under strong PhoQP-activating stress conditions such as exposure to sub-MIC levels of specific antimicrobial peptides, upregulate QueE, which then modulates cell division in addition to Q biosynthesis. (b) Methods used to study the two activities of QueE–APB (N-acryloyl-3-aminophenylboronic acid) gel and northern blotting to detect Q-tRNAs and microscopy to observe filamentation. (c) APB gel-northern blot detecting tRNA$^{Tyr}$ in total RNA samples of *E. coli* wild-type (WT, MG1655) and Δ*queE* (SAM31) cells encoding either an empty vector (pEB52) or wild-type (WT) QueE (pRL03). G-tRNA$^{Tyr}$ = unmodified tRNA$^{Tyr}$, Q-tRNA$^{Tyr}$ = Q-modified tRNA$^{Tyr}$. (d) Representative phase-contrast micrographs of *E. coli* wild-type (WT, MG1655) and Δ*queE* (SAM31) cells expressing either an empty vector (pEB52) or WT QueE (pRL03), scale bar = 5 μm. For (c) and (d), cells were grown in supplemented MinA minimal medium for 2 hours (OD$_{600}$ = 0.2–0.3) and induced with IPTG ("+") for 3 hours.

500 μM, we observe a significant increase in the mean cell length, ranging from approximately 3 to 25 μm, respectively (Fig 2). The average cell length obtained for uninduced cells carrying *queE* on a plasmid is similar to that of the cells containing an empty vector. This indicates that any leaky expression of *queE* encoded by the plasmid has a negligible impact on the cell length. The average cell lengths of 18–25 μm obtained for 250–500 μM of IPTG are comparable to those observed for WT cells grown in the presence of sub-MIC level of cationic antimicrobial peptide or Δ*mgrB* cells starved for Mg$^{2+}$ [10], suggesting that overexpression of QueE on a plasmid is a good proxy for upregulation of this enzyme under strong PhoQP-activating stress

conditions. It is to be noted that the average cell lengths at higher levels of induction are likely underestimated due to the technical limitations in fitting some of the long filaments into the imaging window during microscopy. Overall, our data show a gradual and linear increase in the extent of filamentation with induction. To confirm if higher induction indeed results in increased levels of QueE protein in cells, we cloned and expressed a His$_6$-tagged *Ec*QueE on an IPTG-inducible plasmid. This construct behaves similarly to our untagged QueE in causing filamentation upon induction and allows us to monitor His$_6$-*Ec*QueE levels at different inducer concentrations (Fig (i)a and (i)b in S1 Text). Consistent with our expectation, higher IPTG concentrations correlate with higher amounts of His$_6$-*Ec*QueE in cells as visualized by western blotting (Fig (i)c and (i)d in S1 Text), and average cell lengths correlate with QueE abundance (Fig (i)f in S1 Text). As an added control, we show that the His$_6$-*Ec*QueE maintains the biosynthetic activity to produce Q-tRNAs (Fig (i)e in S1 Text). Together, our data indicate that QueE levels need to be about 3-fold or higher than basal (uninduced) expression level to cause cell elongation by greater than 2-fold. Induction with IPTG at 250–500 µM (average cell lengths ranging from 18–25 µm) leads to an estimated 7 to 8-fold increase in QueE levels. In filaments produced under PhoQP activation conditions (WT cells grown in the presence of sub-MIC level of cationic antimicrobial peptide or *ΔmgrB* cells starved for Mg$^{2+}$ [10],), the cell lengths obtained are comparable to plasmid overexpression, where the QueE levels would be upregulated by ~7 fold. Collectively, these results show that QueE levels modulate cell division frequency in a dose-dependent manner.

## QueE's function in queuosine-tRNA biosynthesis is independent of its role as a cell division regulator

Previous genetic analysis has shown that the cellular filamentation phenotype observed during the stress response is QueE-dependent but independent of the other components involved in Q synthesis [10]. We wondered if QueE's role in regulating cell division is functionally linked to its activity as a Q biosynthetic enzyme. In other words, does the catalytic active site of QueE contribute to its ability to inhibit septation? To answer this question, we selected amino acid residues in *Ec*QueE vital for Q synthesis based on the crystal structure [20] and performed alanine scanning mutagenesis (Fig 3A and 3B). Amino acids Q13, E15, and R27 participate in substrate binding, three Cys residues–C31, C35, and C38 bind iron-sulfur clusters, T40 binds magnesium, and residues G94, S136, and Q189 are predicted to be involved in cofactor (S-adenosyl methionine) binding [19,20]. We also included in our analysis three lysine residues identified to be acetylation sites in *Ec*QueE–K60, K66, and K194 as lysine acetylation may affect protein activity [36–39].

   We cloned the selected single alanine mutants of QueE on an IPTG-inducible plasmid and expressed each variant in a Δ*queE* background. We then analyzed the effect of each mutant on Q-tRNA production and cell morphology using APB gel-northern blotting [34,35] and microscopy, respectively (Figs 3 and (ii) in S1 Text). Notably, mutations at several positions believed to be important for catalysis (Q13, T40, G94, S136, and Q189) and acetylation of QueE (K60 and K194) have a negligible effect on either function (Fig (ii) in S1 Text). While an individual mutation may affect the catalytic speed or efficiency of QueE, in this study, we are interested in the overall effect of a mutation in QueE on the formation of the end product (modified Q-tRNA) as a proxy for a functional biosynthetic enzyme. Therefore, these mutations were not considered for further analysis.

   Interestingly, one of the mutants, QueE R27A, abolished Q-tRNA formation but retained the ability to cause filamentation like the WT (Figs 3C, 3D, and (ii) in S1 Text). The C38A mutation also prevents Q-tRNA formation, however, expression of this mutant does not lead

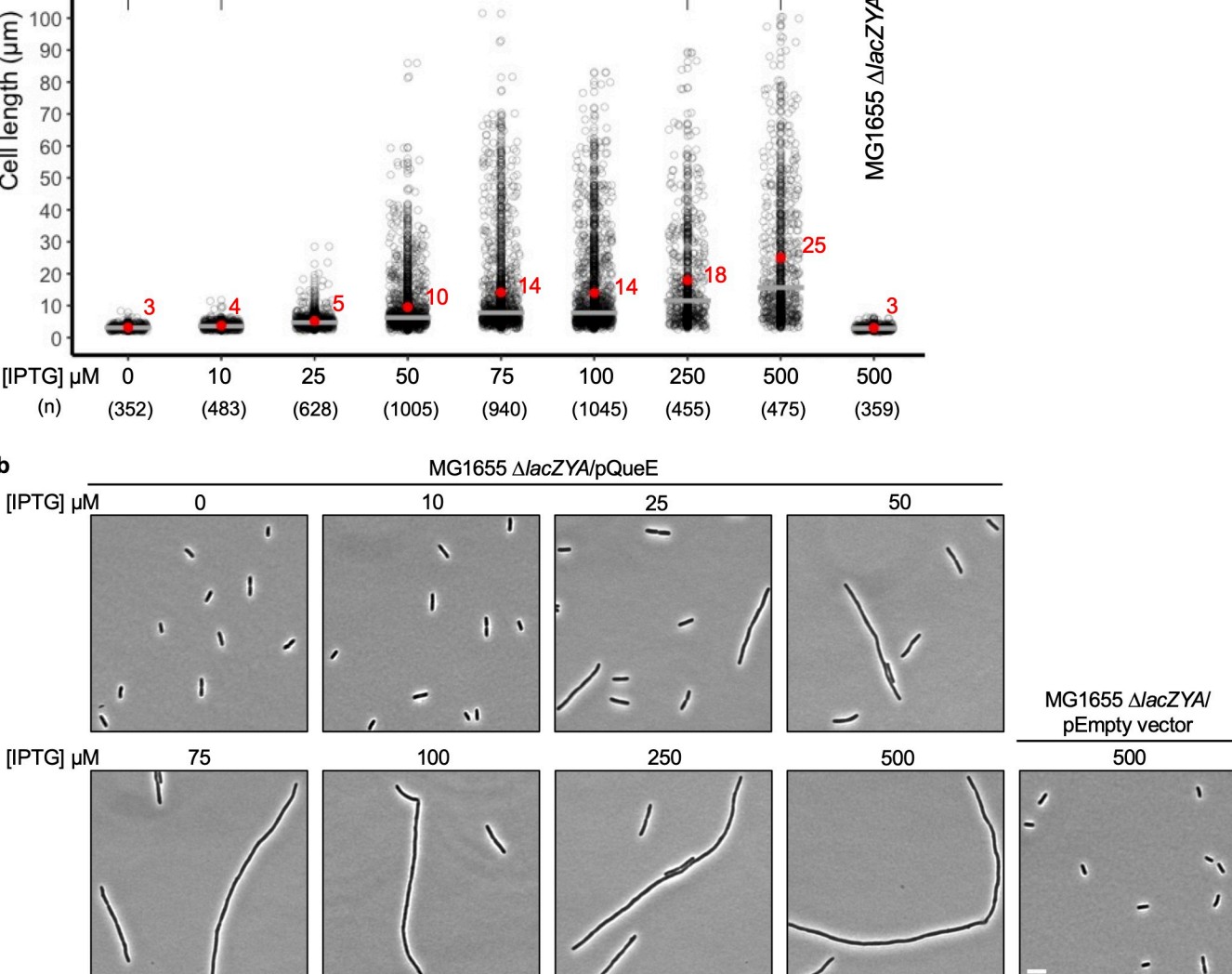

**Fig 2. Effect of dose-dependent expression of *Ec*QueE on septation.** (a) Measurement of cell lengths of *E. coli* MG1655 Δ*lacZYA* (TIM183) cells expressing QueE. Strains containing a plasmid encoding *E. coli* QueE (pRL03) or empty vector (pEB52) were grown in supplemented MinA minimal medium for 2 hours ($OD_{600}$ = 0.2–0.3) and induced with IPTG for 3 hours at the indicated concentration. Gray circles represent individual cells, mean cell length values are indicated in red, and the horizontal gray bars represent the median. Data are obtained from four independent experiments and the number of cells analyzed for each inducer concentration is indicated by (n). Statistical analysis was done using t-test displaying significance at *$P \leq 0.05$, **$P \leq 0.01$, ***$P \leq 0.001$, ****$P \leq 0.0001$, and "ns" = $P > 0.05$ (b) Representative phase-contrast micrographs of Δ*lacZYA* cells expressing QueE (pRL03) at the indicated inducer concentration, scale bar = 5 μm.

to filamentation, suggesting that the mutation may disrupt overall protein structure and stability. To further characterize the cellular localization of the R27A mutant, we created an N-terminally-tagged YFP-QueE-R27A fusion. The YFP-QueE-WT clone behaves similarly to the

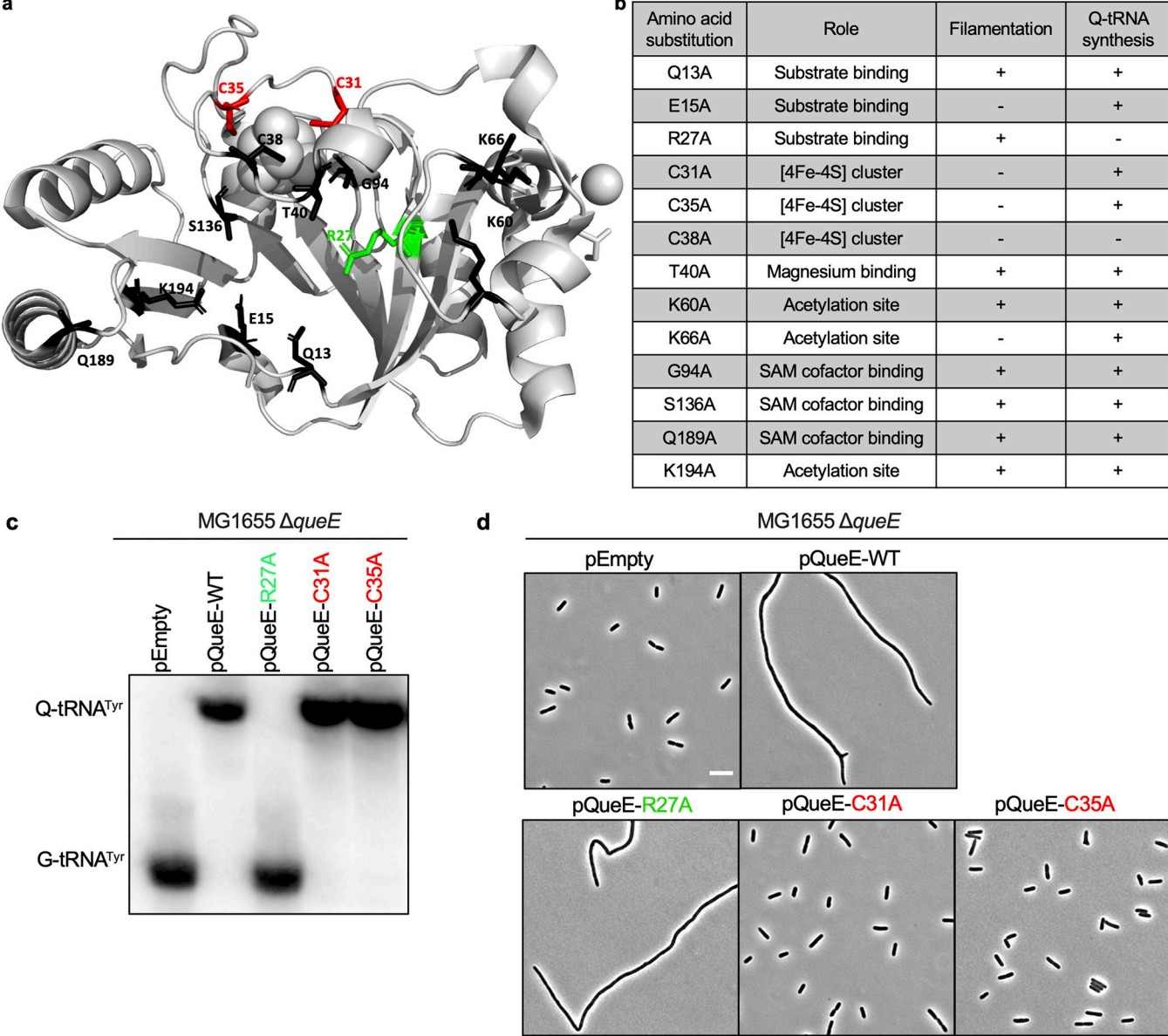

**Fig 3. Effect of single alanine substitutions of conserved amino acid residues in *Ec*QueE on Q biosynthesis and cell division.** (a) Structure of *E. coli* QueE (EcQueE, PDB ID 6NHL) showing amino acid residues, catalytically important for Q biosynthesis. (b) Summary of alanine scanning mutagenesis of key amino acid residues in *Ec*QueE and the effect of each mutant on cellular morphology and Q biosynthesis indicated by (+) or (-). (c) APB gel-northern blot detecting tRNA$^{Tyr}$ in total RNA samples of Δ*queE* (SAM31) cells encoding either WT QueE (pRL03), QueE-R27A (pSY98), QueE-C31A (pSA1), QueE-C35A (pSA2) or an empty vector (pEB52). G-tRNA$^{Tyr}$ = unmodified tRNA$^{Tyr}$, Q-tRNA$^{Tyr}$ = Q-modified tRNA$^{Tyr}$. (d) Representative phase-contrast micrographs of Δ*queE* (SAM31) cells expressing WT QueE and selected mutants, scale bar = 5 μm. For (c) and (d), cells were grown in supplemented MinA minimal medium for 2 hours (OD$_{600}$ = 0.2–0.3) and induced with 0.5 mM IPTG for 3 hours.

untagged protein in modulating septation [10] and is functional in Q-biosynthesis (Fig (ii)d in S1 Text). The R27A construct localizes to the Z-ring similarly to the localization of the WT (Fig 4A,[10]). In addition, the mean length of the filamenting cells observed for the R27A mutant is comparable to that of WT, even though the fluorescence signal for the R27A mutant suggests a somewhat lower level of its expression relative to the WT (Fig 4C and 4D). QueE-mediated filamentation also occurs when cells lacking MgrB–a negative feedback inhibitor of

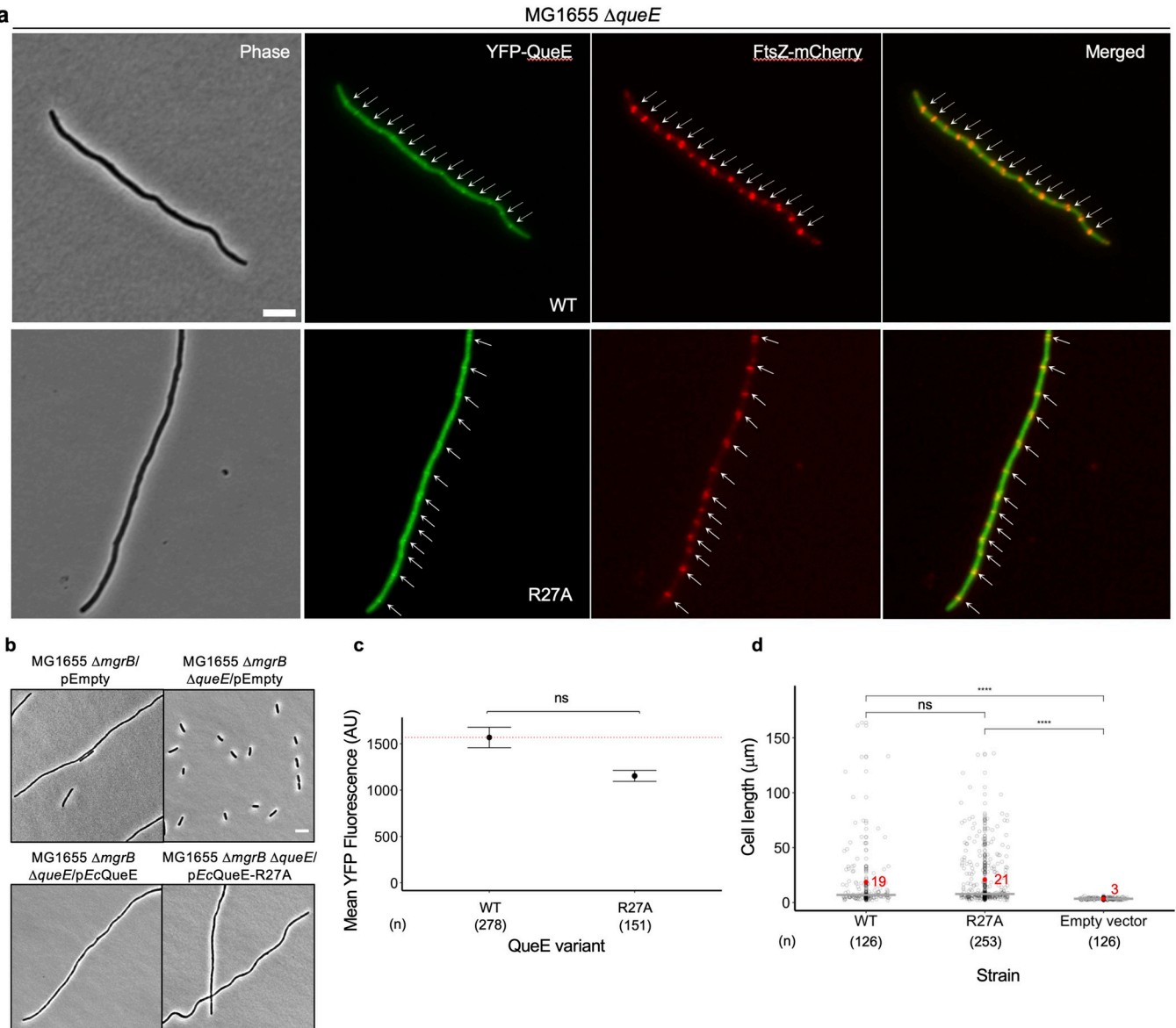

**Fig 4. Role of *Ec*QueE-R27A mutant, deficient in Q biosynthesis, in cell division.** (a) Representative phase-contrast and fluorescence micrographs of *ΔqueE* (SAM96) cells co-expressing FtsZ-mCherry (pEG4) and either YFP-EcQueE (pSY76) or YFP-*Ec*QueE-R27A (pSA58). Cells were grown in supplemented MinA minimal medium for 4 hours, followed by 0.5% arabinose induction for FtsZ-mCherry expression for another 1 hour (please see Methods for details on pSY76 and derivatives). Green and red arrowheads represent the localization of YFP-QueE and FtsZ- mCherry, respectively. (b) Representative phase-contrast micrographs of *ΔmgrB* and *ΔmgrBΔqueE* cells containing either an empty plasmid (pEB52), or plasmids expressing *Ec*QueE (pRL03) and *Ec*QueE-R27A (pSY98), scale bar = 5 μm. Cells were grown in supplemented MinA minimal medium for 2 hours (OD$_{600}$ = 0.2–0.3) and induced with 0.5 mM IPTG for 3 hours. (c) Quantification of YFP-QueE levels in *ΔqueE* (SAM96) cells transformed with plasmids encoding YFP-tagged WT *Ec*QueE (pSY76), *Ec*QueE-R27A (pSA58), or an empty vector (pEB52). Cells were grown in supplemented MinA minimal medium for 5 hours. Data represent the mean and range from two independent experiments and fluorescence was quantified from the number of cells (n) per sample as indicated. The red dotted line represents the expression level for WT YFP-QueE. (d) Measurement of cell lengths of *ΔqueE* (SAM96) cells encoding WT *Ec*QueE (pRL03), *Ec*QueE-R27A (pSY98), or an empty vector (pEB52). Cells were grown in supplemented MinA minimal medium for 2 hours (OD$_{600}$ = 0.2–0.3) and induced with 0.5 mM IPTG for 3 hours. Gray circles represent individual cells, mean cell length values are indicated in red and the horizontal gray bars represent the median. Data are obtained from three independent experiments and the total number of cells (n) per sample is indicated. Statistical analysis was done using t-test displaying significance at *P $\leq$ 0.05, **P $\leq$ 0.01, ***P $\leq$ 0.001, ****P $\leq$ 0.0001, and "ns" = P >0.05.

PhoQ)–are grown in low magnesium conditions [10]. Consistent with our expectation, the QueE-R27A mutant mimics the WT QueE in restoring filamentation in a *ΔmgrB ΔqueE* strain (Fig 4B).

In contrast to the R27A mutant, we identified four mutants, E15A, C31A, C35A, and K66A, which do not affect Q-tRNA synthesis but selectively impair the ability to inhibit septation. To cause a block in cell division, QueE needs to be expressed at high levels ($\geq$ 3X basal levels), so it is possible that these mutants suppressing filamentation are expressed at levels insufficient to hinder division. To determine the expression levels of these individual mutants, we measured fluorescence by incorporating each mutation into the YFP-tagged QueE background. YFP-QueE-E15A and YFP-QueE-K66A show very low fluorescence, indicating that these mutations affect protein stability. Both mutants, YFP-QueE-C31A and YFP-QueE-C35A, display fluorescence at a level comparable to or higher than YFP-QueE-WT, indicating that the lack of filamentation in strains expressing these mutants is not due to a lower amount of protein in cells (Fig (ii)c in S1 Text). In addition, we performed western blotting to confirm that the proteins YFP-QueE-C31A and YFP-QueE-C35A are produced at full length at levels comparable to that of WT (Fig (iii) in S1 Text). It is important to note that IPTG induction and QueE overexpression are not required for Q-tRNA synthesis (Fig 1C), therefore we tested the possibility that the C31A and C35A variants may be catalytically less active than wild-type QueE by performing APB-northern blots for cells expressing these variants with or without induction. QueE-C31A and QueE-C35A showed Q-tRNA$^{Tyr}$ formation even in the absence of induction (Fig (iv) in S1 Text), indicating that these mutants are fully active in Q-tRNA synthesis. In conclusion, we have identified and described single amino acid residues in *Ec*QueE that specifically affect either its function in Q-tRNA biosynthesis or cell division. Together, our results establish that the role of QueE as a modulator of septation is independent of its function in the Q-biosynthetic pathway.

## Dual functions of QueE are conserved among *E. coli* and related bacteria

Consistent with the fact that *de novo* biosynthesis of Q is a characteristic of most prokaryotes [40], QueE is widely conserved among bacteria [14,40,41]. Although the role of *E. coli* QueE (*Ec*QueE) in Q biosynthesis is well established [20], the homolog from *Bacillus subtilis* (*Bs*QueE) was the first to be characterized in terms of its biochemical properties and detailed catalytic mechanism [18,19,42]. Intriguingly, these two enzymes are less than 40% similar ([10,20], Fig (v) in S1 Text). Accordingly, QueE homologs across bacterial genera appear to be highly divergent in terms of their gene and protein sequences, resulting in distinct clusters based on sequence similarity analysis [20]. QueE protein sequences are increasingly dissimilar as we move farther from enterobacteriaceae within bacterial phylogeny. Therefore, we hypothesized that the sequence divergence among QueE orthologs may impact their functions in Q-tRNA biosynthesis and/or regulation of cell division. To test this hypothesis, we selected and analyzed orthologs of QueE from other gamma proteobacteria–*Klebsiella pneumoniae* (*Kp*QueE), *Salmonella Typhimurium* (StQueE), *Pseudomonas aeruginosa* (*Pa*QueE), and a firmicute, *Bacillus subtilis* (BsQueE). Based on the multiple sequence alignment of selected QueE orthologs and the corresponding phylogenetic tree (Fig (v) in S1 Text), as expected *Bs*QueE is the most distant ortholog, followed by *Pa*QueE.

To test the ability of each QueE ortholog to synthesize Q-tRNAs and inhibit cell division, we cloned each gene into an IPTG inducible multicopy plasmid and induced expression in *E. coli ΔqueE* cells. All of the QueE orthologs tested here are functional in producing tRNAs modified with Q (Fig 5A), indicating that this function is well-conserved among these orthologs. However, when we observe the morphology of cells expressing these orthologs, interestingly,

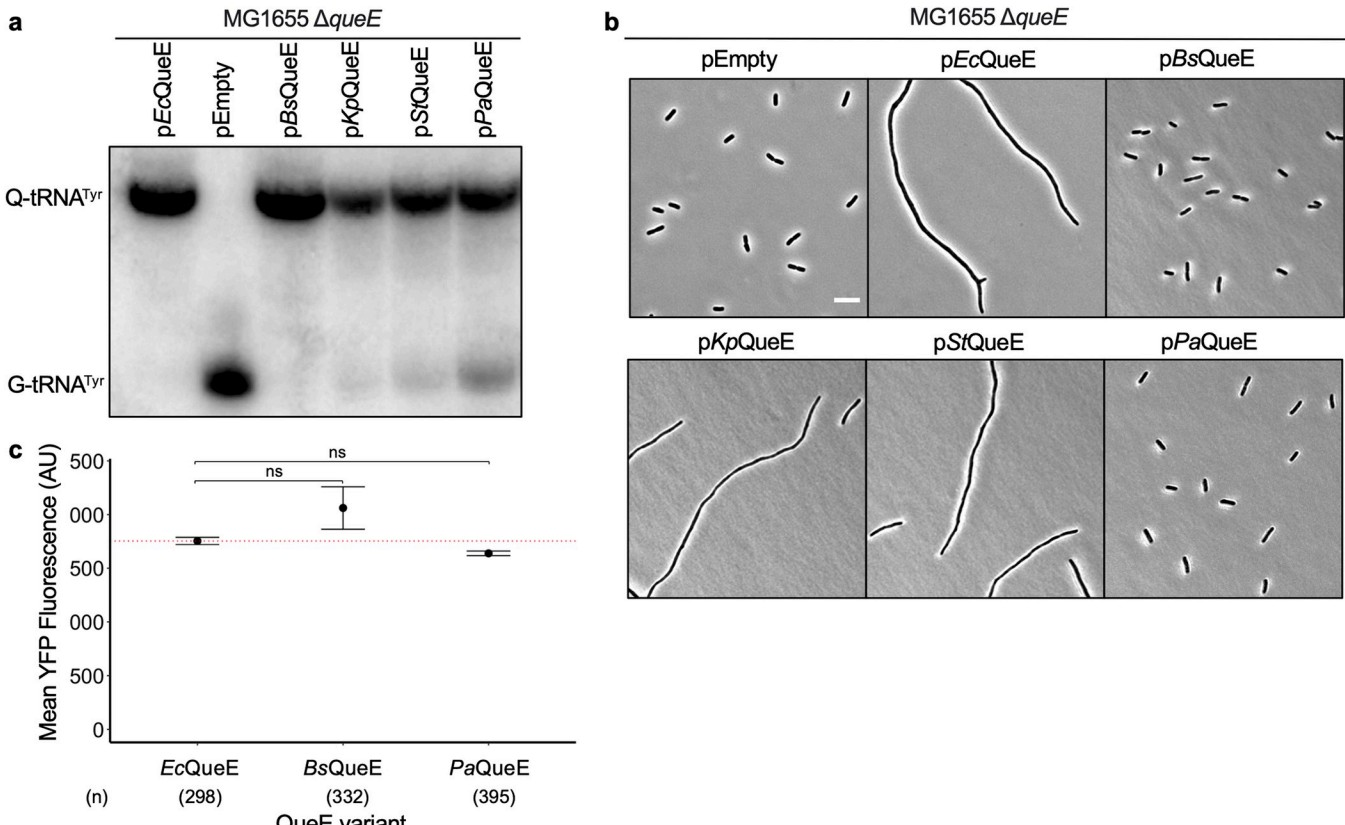

**Fig 5. Analysis of dual functions of QueE among orthologs from *E. coli* and closely related genera.** (a) APB gel-northern blot detecting tRNA$^{Tyr}$ in total RNA samples of Δ*queE* (SAM31) cells encoding QueE orthologs from *E. coli* (*Ec*QueE, pRL03), *Bacillus subtilis* (*Bs*QueE, pSY45), *Klebsiella pneumoniae* (*Kp*QueE, pSA5), *Salmonella Typhimurium* (*St*QueE, pSA21), *Pseudomonas aeruginosa* (*Pa*QueE, pSA127) and an empty vector (pEB52). G-tRNA$^{Tyr}$ = unmodified tRNA$^{Tyr}$, Q-tRNA$^{Tyr}$ = Q-modified tRNA$^{Tyr}$. (b) Representative phase-contrast micrographs of Δ*queE* (SAM31) cells transformed with plasmids encoding either *Ec*QueE, *Bs*QueE, *Kp*QueE, *St*QueE, p*Pa*QueE, or an empty vector, scale bar = 5μm. For (a) and (b), cells were grown in supplemented MinA minimal medium for 2 hours (OD$_{600}$ = 0.2–0.3) and induced with 0.5 mM IPTG for 3 hours. (c) Quantification of YFP-QueE levels in *E. coli* Δ*queE* (SAM96) cells transformed with plasmids encoding YFP-tagged *Ec*QueE (pSY76), *Bs*QueE (pSA121), *Pa*QueE (pSA126), or an empty vector (pEB52). Cells were grown in supplemented MinA minimal medium for 5 hours (please see Methods for details on pSY76 and derivatives). Data represent the mean and range from two independent experiments, and fluorescence was quantified from the number of cells (n) per sample as indicated. The red dotted line represents the expression level for WT YFP-QueE, Statistical analysis was done using t-test displaying significance at *P ≤ 0.05, **P ≤ 0.01, ***P ≤ 0.001, ****P ≤ 0.0001, and "ns" = P >0.05.

only *Kp*QueE and *St*QueE trigger inhibition of septation, but *Bs*QueE and *Pa*QueE do not, when expressed in *E. coli* Δ*queE* cells (Fig 5B). We considered the possibility that heterologous expression of *Bs*QueE and *Pa*QueE may affect their folding and stability. So, we analyzed the expression of these two QueE orthologs using the YFP-tagged protein fusions along with *Ec*QueE as a control. Upon expression, both YFP-*Bs*QueE and YFP-*Pa*QueE show similar levels of fluorescence as that of YFP-*Ec*QueE (Fig 5C) and we confirmed that the proteins YFP-*Bs*QueE and YFP-*Pa*QueE are produced at full length at levels comparable to that of YFP-*Ec*QueE by western blotting (Fig (iii) in S1 Text).

While the expression of orthologous proteins in *E. coli* is convenient, it is not testing their function in the native environment where additional factors, which may help cause filamentation, are missing. To test this scenario, we expressed each QueE ortholog on a plasmid in its original host, i.e., *Kp*QueE in *K. pneumoniae*, *Pa*QueE in *P. aeruginosa* cells, and so on. For stable expression of *Bs*QueE in *B. subtilis*, we prepared a genomic construct carrying an IPTG-inducible *Bs*QueE at the *amyE* locus (details in the methods section). As observed for

expression in *E. coli*, *Kp*QueE and *St*QueE cause filamentation in their corresponding host cells, however, *Bs*QueE and *Pa*QueE fail to do so (Fig (vi) in S1 Text). The average cell lengths for *P. aeruginosa* cells expressing *Pa*QueE were similar to that of control cells carrying an empty vector (Fig (vi)b in S1 Text). Notably, *B. subtilis* cells show cell elongation and/or chaining to varying extents irrespective of QueE expression but there is no change in average cell lengths with increased QueE expression. However, there is still a possibility that partial or complete septa may be formed between cells within a chain of *B. subtilis* cells expressing *Bs*QueE. We stained the cytoplasmic membrane with the FM4-64 dye to visualize any additional invaginations or septa *B. subtilis* cells expressing *Bs*QueE vs. control cells that do not express this protein. We did not observe significant differences in septa formed between the two cell types, indicating that *Bs*QueE may not impact cell division in *B. subtilis* (Fig (vi)c in S1 Text). To ensure that there is no problem with overexpression of either *Pa*QueE or *Bs*QueE in their respective hosts, we generated YFP-tagged versions of these proteins and measured YFP fluorescence as a function of their level of expression with and without induction. Both YFP-*Pa*QueE and YFP-*Bs*QueE show a robust increase in YFP fluorescence upon induction, and we confirmed the corresponding expression of the full-length protein by western blotting (Fig (vi)d and (vi)e in S1 Text). There were no bands observed for uninduced samples, suggesting a tight regulation of protein expression in these constructs. Despite the strong induction and expression of *Bs*QueE in *B. subtilis* and *Pa*QueE in *P. aeruginosa* the impact on cell division, if any, seems negligible. Taken together, these results show that the ability of QueE to disrupt cell division is specific to enterobacterial counterparts, closely related to that of *E. coli*.

## A distinct region (E45-E51) in *E. coli* QueE is dispensable for its role in queuosine-tRNA biosynthesis but not for the regulation of septation

Besides sequence variation, the QueE enzyme family also displays a high degree of structural divergence [20], suggesting a molecular basis for the differences observed in the ability of QueE orthologs to regulate cell division. Among the selected QueE proteins, *Bs*QueE and *Pa*QueE do not cause filamentation. *Bs*QueE and *Pa*QueE protein sequences share an identity of only ~19%, 24%, respectively, and <40% similarity with *Ec*QueE (based on EMBOSS Needle pairwise sequence alignments [43], yet they are catalytically active in Q-tRNA synthesis (Figs 5 and (vi) in S1 Text). To gain insights into the mechanism, we generated a structural alignment to compare the structures of *Ec*QueE[20], *Bs*QueE[19], and an AlphaFold [44,45] model of *Pa*QueE (Fig 6A). We see that the catalytic core region corresponding to QueE's CDG synthase activity is preserved between the three QueE proteins despite low sequence similarity Fig 6, [20]. Unsurprisingly, amino acid residues involved in substrate and co-factor binding are well-conserved in the orthologs (Fig (v)a in S1 Text). Based on the structural alignment, we noted a distinguishing feature of *Ec*QueE in the region spanning 22 amino acids E45 through W67 (Fig 6). According to the *Ec*QueE crystal structure, this region comprises two helical motifs– αT2 and α2′, connected to each other and the rest of QueE *via* flexible loops [20]. This E45-W67 region from *Ec*QueE appears to be variable among other orthologs, where it is truncated or almost entirely missing in *Bs*QueE and *Pa*QueE (Fig 6A). We hypothesized that this region with the two helices unique to *Ec*QueE contributes to its second function in inhibiting septation yet dispensable for its Q-biosynthetic function. To test this idea, we cloned *Ec*QueE variants lacking regions comprising either of the helices αT2 (ΔE45-E51), α2′ (ΔV52-W67), or both (ΔE45-W67) and expressed them in Δ*queE* cells. Remarkably, *Ec*QueE mutants carrying deletions of either αT2, α2′, or the entire region E45-W67 are functional in Q-tRNA production (Fig 6B), indicating that the E45-W67 region is dispensable for Q biosynthesis. To check if the deletion mutants are catalytically less active than the WT in Q-tRNA synthesis, we

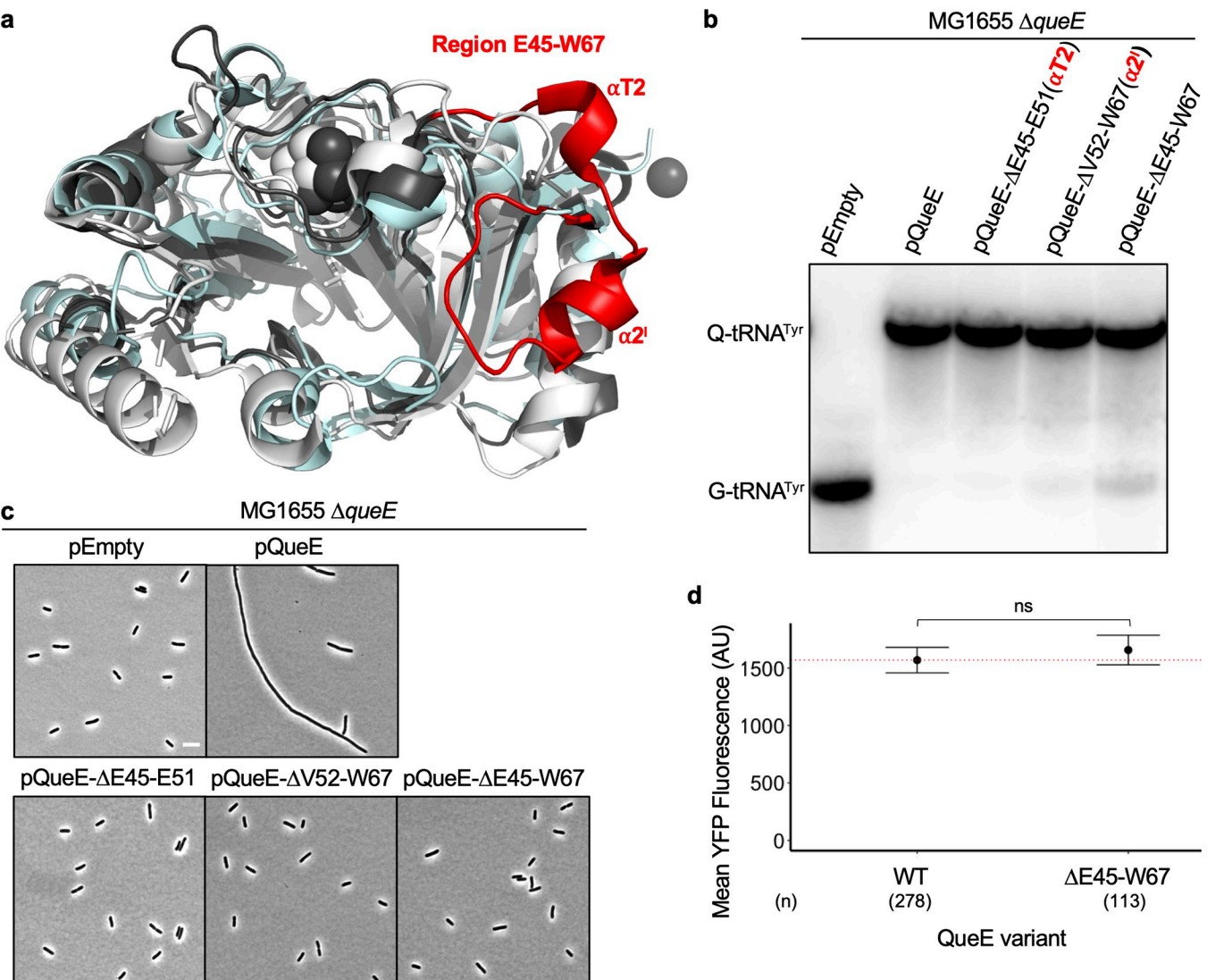

**Fig 6. Role of region E45-W67 in *E. coli* QueE in the biosynthesis of Q-tRNA modification.** (a) Structural alignment of *E. coli* QueE (*Ec*QueE, PDB 6NHL, dark gray), *Bacillus subtilis* QueE (*Bs*QueE, PDB 5TH5, light gray), and an Alphafold-generated model of *P. aeruginosa* QueE (*Pa*QueE, cyan). The variable region E45-W67 in *Ec*QueE is highlighted in red. (b) APB gel-northern blot showing Q-tRNA profiles of *ΔqueE* cells (SAM31) containing IPTG inducible plasmids encoding *E. coli* QueE (pRL03), control vector (pEB52), and selected mutants of *Ec*QueE—pQueE-ΔE45-E51 (pSA60), pQueE-ΔV52-W67 (pSA61), and pQueE-ΔE45-W67 (pSA62). G-tRNA$^{Tyr}$ = unmodified tRNA$^{Tyr}$, Q-tRNA$^{Tyr}$ = Q-modified tRNA$^{Tyr}$ (c) Phase contrast micrographs of *ΔqueE* (SAM31) cells transformed with an empty vector and plasmids encoding Wildtype (WT) *Ec*QueE and the selected *Ec*QueE peptide deletion mutants (as in (a) above), Scale bar = 5 μm. For (b) and (c), cells were grown in supplemented MinA minimal medium for 2 hours (OD$_{600}$ = 0.2–0.3) and induced with 0.5 mM IPTG for 3 hours. (d) Quantification of QueE levels in *ΔqueE* (SAM96) cells transformed with an empty vector and IPTG inducible plasmids encoding YFP-tagged wild-type QueE (WT, pSY76) and QueE ΔE45-W67 (pSA76). Cells were grown in supplemented MinA minimal medium for 5 hours (please see Methods for details on pSY76 and derivatives). Dots and the error bars represent the mean and range from two independent experiments with quantified fluorescence, and the total number of cells (n) per sample is indicated. The red dotted line represents the expression levels for WT. Statistical analysis was done using t-test displaying significance at *P $\leq$ 0.05, **P $\leq$ 0.01, ***P $\leq$ 0.001, ****P $\leq$ 0.0001, and "ns" = P >0.05.

performed APB-northern blotting for cells expressing these deletion mutants with or without induction. We noticed that the *Ec*QueE ΔE45-E51 mutant is fully functional in forming Q-tRNA$^{Tyr}$ in cells that are uninduced or induced with IPTG (Figs 6B and (vii) in S1 Text). The other two deletions (ΔV52-W67 and ΔE45-W67) do not show significant Q-tRNA$^{Tyr}$ in the absence of induction. When the cells are induced with IPTG both these mutants form Q-

tRNA$^{Tyr}$ albeit to a lower extent relative to the *Ec*QueE ΔE45-E51 mutant and the WT, suggesting that the ΔV52-W67 and ΔE45-W67 mutants are catalytically weaker than the WT. It is notable the ΔV52-W67 mutant only has a slight reduction in Q-tRNA formation and the ΔE45-W67 mutant still retains the ability to form Q-tRNA upon overexpression (Figs 6B and (vii) in S1 Text), a condition that mimics PhoQP-activating stress where QueE is upregulated. Then, we observed the cell morphology and none of the deletion constructs caused filamentation (Fig 6C), supporting our hypothesis that the E45-W67 region mediates QueE's role in cell division regulation. To control for the possibility that the deletion of this region affects overall protein stability and level in the cell, we quantified protein levels using a YFP-QueE-ΔE45-W67 construct by monitoring cell fluorescence. We see that the level of fluorescence for YFP-QueE-ΔE45-W67 is comparable to that of WT (Fig 6D) and we confirmed that the full-length protein YFP-QueE-ΔE45-W67 is produced at levels comparable to that of YFP-QueE WT by western blotting (Fig (iii) in S1 Text).

Given the ability to cause filamentation by *St*QueE and *Kp*QueE, closely related orthologs to *Ec*QueE, we wondered if there is a consensus sequence that can predict whether a given QueE sequence may confer an ability to inhibit cell division. We identify several conserved residues using bioinformatic analysis of QueE protein sequences from 18 representative enterobacteria (Fig (viii) in S1 Text). To better represent the amino acid diversity at each position, we generated a WebLogo [46] corresponding to the E45-W67 region in *Ec*QueE, revealing significant features of this sequence (Fig 7). In particular, amino acids Glu or Asp at position 45, Ile or Val at position 57, Lys60, Thr61, Glu or Asp at position 63 and Trp67 (based on *Ec*QueE numbering) are highly conserved suggesting that their occurrence correlates with QueE's ability to impair septation.

## Discussion

Moonlighting proteins highlight the versatility and adaptability of bacterial proteins [1,2]. Studying these multifunctional proteins in a model organism such as *E. coli* provides a deeper understanding of protein functionality. Bacterial cell division is a complex process orchestrated primarily by the essential proteins that form the divisome, with FtsZ being one of the most central and well-studied [26,27,47–51]. However, in addition to these core proteins,

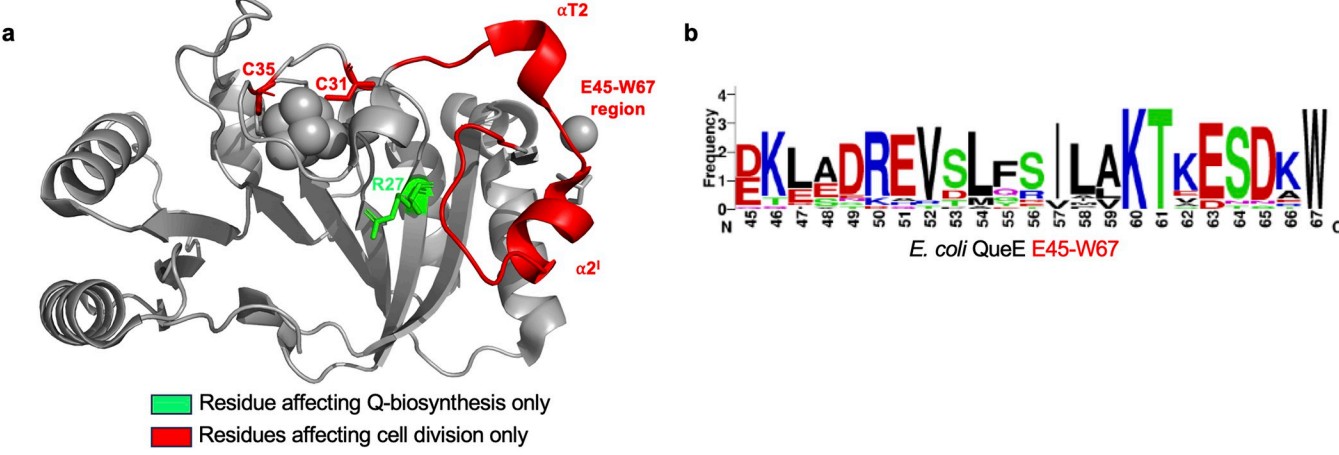

**Fig 7. QueE's dual roles in Q-biosynthesis and septation inhibition.** (a) Structure of *E. coli* QueE (PDB 6NHL) showing residue affecting Q-biosynthesis only (green, R27) and septation inhibition only (red, C31, C35, E45-W67). (b) A consensus sequence of the E45-W67 region from 18 representative enterobacteria (see Fig (viii) in S1 Text for details) was generated using WebLogo [71].

there are non-essential, auxiliary proteins that, while not strictly required for cell division, play supportive roles in regulating the process. A notable example is SulA, an inhibitor of FtsZ polymerization induced in response to DNA damage to temporarily halt cell division [49,52–54]. Accessory proteins, SlmA and Noc, ensure division occurs at appropriate sites and not over the chromosome [55–58]. These and several other auxiliary proteins, while "non-essential", can be crucial for survival to fine-tune and ensure optimal cell division under a variety of conditions in natural environments [59–61].

The enzyme QueE, known for its role in Q biosynthesis [62], also regulates septation [10]. While QueE-dependent filamentation is not linked to the Q-biosynthetic pathway, it was unclear how QueE performs these two unrelated functions. Here, we show that the regulation of cell size is dependent on the levels of QueE. We find that as QueE expression increases, the average cell length also increases. Using site-directed mutagenesis of catalytically important amino acid residues in *Ec*QueE [20], we identify specific residues that selectively affect filamentation or Q-biosynthesis. In particular, the QueE-R27A mutant is catalytically dead for Q-biosynthesis yet modulates septation. Like the WT, QueE-R27A localizes with FtsZ and the Z-ring, indicating that the septal localization is still maintained in this mutant that is catalytically inactive for Q-biosynthesis. Thus, QueE's role in queuosine biosynthesis and modulation of cell division are not functionally linked, elucidating QueE as a moonlighting protein (Fig 7).

Based on the structural alignment of QueE orthologs, we observe that the catalytic core domain required for Q biosynthesis is well-preserved (Fig 6,[20]) and the expression of different QueE enzymes in an *E. coli ΔqueE* strain restores Q-tRNA synthesis (Fig 5). However, some peripheral structural elements, especially the regions surrounding the iron-sulfur cluster binding pocket, show remarkable variation among members of the QueE family ([20], Figs 6 and (v)a in S1 Text). Our results are consistent with the previous analysis that the sequence and structural diversity among QueE proteins do not impact its catalytic function or cofactor binding. Interestingly, Grell and colleagues ponder over the significance of such large variation in QueE structures [20]. They show that differences in surface elements of QueE as a radical SAM enzyme are at least in part a likely outcome of co-evolution with the cognate flavodoxin that allows for efficient protein-protein interactions [20]. Here, we note a stark variation in the region encompassing the two helices α2′ and αT2 based on our structural comparison of *Ec*QueE, *Bs*QueE and *Pa*QueE (AlphaFold-generated model). We propose these variable features may explain whether or not a given QueE protein has a moonlighting function in modulating cell division. A particular region of interest–comprising E45 through W67 is distinctive to *Ec*QueE, and a sub-region E45 through E51 (αT2) is dispensable for Q biosynthesis but required to cause filamentation (Fig 6). Specific amino acids in this region are highly conserved among enterobacterial genera (Fig 7) and potentially mediate the interactions between QueE and proteins at the Z-ring during cell division. We demonstrate that in bacteria from at least three distinct genera, elevated expression of the native QueE ortholog leads to filamentous cell growth (Fig 5). This indicates that such a mechanism for cell division regulation is widely conserved among Enterobacteriaceae. Two other distant homologs from *P. aeruginosa* and *B. subtilis*, exhibiting lower sequence and structural similarity to *Ec*QueE than enterobacterial orthologs, do not show functional conservation in regulating septation. It is tempting to speculate that the lack of cell division regulation is primarily due to the sequence and structural differences between the QueE orthologs. If this is the case, it would be interesting to see if *Ec*QueE expression in *P. aeruginosa* can cause filamentation. To test this idea, we expressed *Ec*QueE on a shuttle vector in both *E. coli* and *P. aeruginosa*. Interestingly, we do not see any filamentation in *P. aeruginosa* but our control *E. coli* cells displayed filaments (Fig (ix) in S1 Text), indicating that there are likely additional factors and variations in cell division machinery in Pseudomonas and Bacillus involved in modulating cell division. The target(s) that QueE

binds to and the mechanism by which it controls septation in *E. coli* are subjects of an ongoing inquiry.

Overall, bacteria possess diverse, species-specific regulators that modulate physiological responses based on environmental conditions or the cell's metabolic state. QueE serves as an example to underscore the intricate ways in which bacteria have evolved to use proteins in multiple roles, ensuring efficient coordination between processes like cell division and other essential cellular functions.

## Materials and methods

### Strains and plasmids

Details of the strains, plasmids, and oligonucleotides used in this study are provided in the supporting information (Tables A, B, and C, respectively, in S1 Text). *E. coli* K-12 MG1655 strain and its derivatives, plasmid pEB52, and its derivatives were gifts from Dr. Mark Goulian (University of Pennsylvania). The following strains, *Pseudomonas aeruginosa* PAO1, *Klebsiella pneumoniae* ATCC 13883, *Bacillus subtilis* 168, *Bacillus subtilis* IS75, and *S. Typhimurium* 14028, were gifts from Drs. Bryce Nickels (Rutgers), Huizhou Fan (Rutgers), Jeffrey Boyd (Rutgers), David Dubnau (Rutgers), and Dieter Schifferli (University of Pennsylvania); plasmids pEG4 and pPSV38 were gifts from Drs. Kenn Gerdes (Copenhagen) and Simon Dove (Boston Children's Hospital, Harvard Medical School), respectively.

### Cloning of QueE and its variants

Plasmids encoding single alanine mutants of *E. coli* QueE, His$_6$-QueE, and YFP-QueE were made using inverse PCR or single primer methods using pRL03 (untagged QueE), pSY85 (His$_6$-QueE) or pSY76 (YFP-QueE) as templates. All the plasmids used to test QueE expression in *E. coli* are derivatives of pTrc99a [63]. pTrc plasmids are IPTG-inducible and considered medium copy with a copy number of ~30 per cell and they carry *lacI$^q$*, which represses the lac promoter reducing leaky expression. QueE orthologs from *S. Typhimurium* 14028 and *Pseudomonas aeruginosa* PAO1 were cloned using restriction sites, EcoRI and BamHI, while *Bacillus subtilis* QueE was cloned using the SacI and BamHI restriction sites in a pEB52 vector resulting in the plasmids pSA21, pSA22, and pSY45 respectively. Since *K. pneumoniae* ATCC 13883 has an intrinsic resistance to ampicillin/carbenicillin, the QueE gene from *K. pneumoniae* was cloned into a pBAD33 vector using the Xba1 and Kpn1 restriction sites to create plasmid pSA59. We also added a ribosome binding site (RBS) to pBAD33, upstream of the start site of the cloned gene using the inverse PCR cloning method. Plasmids pSA60, pSA61, and pSA62 encoding variants of *E. coli* QueE, QueE-ΔE45-E51, QueE-ΔV52-W67, and QueE-ΔE45-W67 were made using inverse PCR. YFP-tagged QueE variants (pSA57-58, pSA73, pSA75-76, pSA119) were created by inverse PCR method, YFP-*Bs*QueE (pSA121) and YFP-*Pa*QueE (pSA126) for expression in *E. coli* were made using the NEB Hi-Fi Assembly protocol by substituting *Ec*QueE in the pSY76 background. Plasmid pSY76 (described in [10]) is a pTrc99a derivative containing a strong RBS (AAAGAGGAGA) at an optimal distance of 8 bp upstream of *yfp-queE*, which does not require the addition of IPTG for overexpression in *E. coli*. To prepare plasmid constructs for expression in *P. aeruginosa*, we used the shuttle vector pSV38 [64]. *Pa*QueE, YFP-*Pa*QueE, and *Ec*QueE were amplified from pSA151, pSA126, and pRL03, respectively, using the sequencing primers. The fragments were cloned into the pPSV38 vector using the Xba1 and EcoRI, and XbaI and HindIII, and EcoR1 and XbaI, cut sites to obtain pSA154, pSA156, and pSA157 respectively. Plasmids pSA154, pSA156, and pSA157 were transformed into *P. aeruginosa* PAO1 and selected on LB agar containing 30 μg mL$^{-1}$ gentamycin, while pSA157 was transformed into *E. coli* MG1655 and selected on 15 μg mL$^{-1}$ gentamycin.

All constructs were confirmed by Sanger sequencing, and plasmids were transformed into chemically competent *E. coli* Top10 or XL1-Blue strains for maintenance in 20% glycerol at -80˚C. Plasmids were transformed into MG1655 and its derivatives (TIM183, SAM31, or SAM96) as applicable.

## Construction of *B. subtilis* strains encoding *Bs*QueE and YFP-*Bs*QueE

To generate genomic constructs encoding *Bs*QueE and YFP-*Bs*QueE in *B. subtilis*, YFP-tagged and untagged variants of *Bs*QueE were cloned into the pDR111 hyper-SPANK vector using the Sal1 and Nhe1 cut sites to obtain pSA149 and pSA147, respectively, empty vector was set as control. The ribosome binding site AAAGGAGAGGG corresponding to a well-expressed competence gene, *comGA* in *B. subtilis* [65,66] was included upstream of the *Bs*QueE and YFP-*Bs*QueE open reading frames. XL1-Blue cells were transformed with the ligation mixtures and selected using carbenicillin selection at 100 μg mL$^{-1}$. The resulting clones are pDR111 hyper-SPANK derivatives, incapable of replication in *B. subtilis* but carry front and back sequences of *amyE*, suitable for recombination into the *Bacillus subtilis amyE* locus. The constructs contain Phs, the hyper-SPANK promoter, which is IPTG-inducible. After confirming via Sanger sequencing, the plasmids were transformed into *B. subtilis* IS75 and selected for spectinomycin resistance (100 μg mL$^{-1}$). Colonies that grew were further screened for the absence of amylase activity. Briefly, the amylase test was done by inoculating colonies onto starch plates containing 100 μg mL$^{-1}$ spectinomycin. After 8–12 hours of growth, when the plates were exposed to iodine, a halo zone around the colonies indicated a positive amylase activity. Strains that have undergone recombination (without the halo) are chosen for analysis, SAA50 (*B. subtilis* IS75 *ΔamyE*::pSA147 or *amyE*::*BsqueE*), SAA52 (*B. subtilis* IS75 *ΔamyE*:: pSA149 or *amyE*::*yfp-BsqueE*), and SAA53 (*B. subtilis* IS75 *ΔamyE*::pPDR111 or *amyE*::*empty*).

## Media, reagents, and growth conditions

Routine bacterial growth on solid agar was performed using LB Miller medium (IBI scientific) containing 1.5% bacteriological grade agar (VWR Lifesciences) at 37˚C. Liquid cultures were grown at 37˚C with aeration in either LB miller medium or minimal A medium (MinA) (K$_2$HPO$_4$ (10.5 g l$^{-1}$), KH$_2$PO$_4$ (4.5 g l$^{-1}$), (NH$_4$)$_2$SO$_4$ (10.0 g l$^{-1}$), Na$_3$ citrate.2H$_2$O (0.5 g l$^{-1}$), supplemented with 0.1% casamino acids, 0.2% glucose and 1 mM concentration of MgSO$_4$ unless otherwise indicated. Throughout this study, MinA supplemented, as indicated above, will be referred to as supplemented MinA minimal medium. Antibiotics were used at final concentrations of 100 μg mL$^{-1}$ (LB) or 50 μg mL$^{-1}$ (minimal medium) for carbenicillin, 25 μg mL$^{-1}$ for chloramphenicol, 100 μg mL$^{-1}$ for spectinomycin, and 7.5–30 μg mL$^{-1}$ for gentamycin as indicated. The *lac/trc* and *araBAD* promoters were induced using 500 μM b-isopropyl-D-thiogalactoside (IPTG) and 0.5% arabinose, respectively, unless otherwise specified.

For microscopy experiments, overnight cultures of strains harboring specified plasmids were grown in minimal medium (supplemented MinA as described above) and appropriate antibiotics, then back diluted (1:500) into fresh medium unless otherwise indicated. About 4–6 μL of cells were immobilized on glass slides containing agarose pads made from 1% agarose in 1X MinA salts. For the phase contrast microscopy, back-diluted cells were grown for 2 hours (OD$_{600}$ ~0.2) and then induced for 3 hours with either 0.5 mM IPTG or 0.5% arabinose. Cells were grown under the same conditions to detect modified and unmodified tRNA using APB northern blotting. For the fluorescence microscopy and protein localization, SAM96 (*ΔqueE*) cells expressing either YFP-QueE (pSY76) or YFP-QueE-R27A (pSA58) and FtsZ-mCherry (pEG4) into SAM96 were grown for 4 hours, followed by addition of arabinose and growth for another 1 hour to reach OD$_{600}$ ~0.4–0.5. To monitor single-cell fluorescence in

cells expressing QueE and its variants, SAM96 cells containing plasmids (pSA57, pSA58, pSY76, pSA73, pSA75, pSA76, pSA119, pSA121, pSA126) were grown for 5 hours to reach $OD_{600}$ 0.4–0.5. In fluorescence quantification experiments, the cultures were rapidly cooled in an ice slurry, and streptomycin was added at 250 μg mL$^{-1}$ to halt translation. For IPTG titration experiments, overnight cultures of TIM183/pRL03 or TIM183/pSY85 and TIM183/pTra99a were back diluted (1:500) into fresh medium. These cells were grown for 2 hours and then induced with IPTG at concentrations 0, 10, 25, 50, 75, 100, 250, and 500 μM for 3 hours, followed by imaging.

For QueE expression in *P. aeruginosa*, overnight cultures of the *P. aeruginosa* carrying plasmids pSA154, pSA156, and pPSV38 were grown in LB containing 30 μg mL$^{-1}$ gentamycin. *E. coli* cells containing pSA157 and pPSV38 were included as controls and grown in LB containing 15 μg mL$^{-1}$ gentamycin. *P. aeruginosa* cultures were back diluted 1:500 in supplemented MinA containing 20 μg mL$^{-1}$ gentamycin and induced with 2 mM IPTG for 6 hours. For *E. coli* cultures, 7.5 μg mL$^{-1}$ gentamycin and 0.5 mM IPTG were used. 5 μl aliquots were used for microscopy, and the rest were pelleted for Coomassie staining and western blotting.

For QueE expression in *B. subtilis*, overnight cultures of strains SAA50, SAA52, and SAA53 were grown in supplemented MinA containing 100 μg mL$^{-1}$ spectinomycin. The cultures were back diluted 1:500 in fresh medium and induced with 1 mM IPTG for 6 hours. 5 μl aliquots were used for microscopy and the rest were pelleted for Coomassie staining and western blotting. For cytoplasmic membrane staining with FM4-64, cells were grown in supplemented MinA containing 100 μg mL$^{-1}$ spectinomycin and 1 mM IPTG for 5 hours and stained as described in the microscopy section.

## Microscopy and image analysis

Phase contrast and fluorescence microscopy of MG1655 *ΔqueE* (SAM96 or SAM31) cells harboring plasmids of QueE, single alanine mutants, QueE orthologs, and their variants were performed as previously described [65]. 4–6 μL of the cells were immobilized on 1% agarose pads, and their morphology was observed using a Nikon TiE fluorescent microscope with a TI2-S-HU attachable mechanical stage. The images were captured using a Teledyne Photometrics Prime 95B sCMOS camera with 1x1 binning. The Nikon Ti-E's perfect focus system (PFS) ensured continuous focus during imaging. YFP and mCherry fluorescent images were taken with a 100 ms exposure time at 20% intensity, while phase-contrast images were captured with a 40 ms exposure time and 20% intensity. All image acquisition during experiments was managed using Metamorph software version 7.10.3.279. The background fluorescence was determined using the MG1655 *ΔqueE* (SAM96) strain grown under the same conditions. The average cell fluorescence was quantified using ImageJ [67] and the MicrobeJ plugin [68]. Data from independent replicates was plotted using R. For FM4-64 staining, *B. subtilis* cells were treated with FM4-64 (Invitrogen) at a final concentration of 0.1 ug/ml, incubated in the dark for one hour followed by imaging using the mCherry channel.

## APB gel electrophoresis and northern blotting

tRNA$^{Tyr}$ modified with queuosine (Q) in total cell RNA was detected using previously published protocols [67] with the following modifications. Total RNA was extracted from cell pellets using the Trizol RNA extraction method [69]. 150 ng of total RNA was deacylated in 100 mM Tris-HCl (pH 9.0) at 37˚C for 30 minutes. The deacylated RNA was run on a 0.4 mm thick 7% Urea denaturing polyacrylamide gel, supplemented with 0.5% 3-acrylamidophenylboronic acid (APB). The RNA was transferred using a semi-dry system to a Nytran SuPerCharge nylon blotting membrane (VWR). After transfer, the membrane was crosslinked twice

using a UV crosslinker (Fisher Scientific) at optimal conditions (254 nm wavelength and 1200 mJ energy) and hybridized in 50 mL hybridization buffer (750 mM NaCl and 75 mM sodium citrate, 20 mM $Na_2HPO_4$, 7% SDS, 0.04% BSA fraction V (high purity), 0.04% Ficoll 400, and 0.04% polyvinylpyrrolidone) at pH 7.0, supplemented with 200 μL sheared salmon sperm DNA (10 mg/mL). Following hybridization for 1 hour, $5'$-$^{32}$P labeled oligonucleotide (SA1, Table C in S1 Text) with complementarity to tRNA$^{Tyr}$ were added and incubated for 16–18 hours at 50˚C. The blot was washed thoroughly with a low-stringent wash solution (450 mM NaCl and 45 mM sodium citrate, 25 mM $Na_2HPO_4$ (pH 7.2), 5% SDS, 0.2% BSA fraction V (high purity), 0.2% Ficoll 400, 0.2% polyvinylpyrrolidone, and DEPC dd$H_2O$) at 50˚C for 15 minutes each, and once with stringent wash solution (150 mM NaCl and 15 mM sodium citrate, 1% SDS, DEPC dd$H_2O$) at 50˚C for 5 minutes. The blot was exposed to a phosphor imager screen and imaged using the phosphorimager (GE Amersham Typhoon) at a sensitivity of 4000.

### Cell lysate preparation and western blotting

Cell pellets harvested from 25 mL cultures of TIM183/pSY85 grown in the absence or presence of varying concentrations of inducer (IPTG) were resuspended in 400 μL 20% sucrose/30 mM Tris pH 8.0. and 1 mL 3mM EDTA pH 7.2 was added. The cells were sonicated for ~30 sec (10-sec pulse, 10-sec gap, 3x) using a sonicator (Fisher Scientific, VCX-130 Vibra-Cell Ultrasonic). The resulting lysed cells were centrifuged at 600rpm for 5 minutes, and the supernatant was collected for further use. Each sample is normalized by the cellular biomass and loaded onto two 12% Bis tris gels using MES as a running buffer at 160V for 1 hour. One gel is stained using the Coomassie gel protocol, destained, and imaged using a Biorad Gel Doc XR+ System, while the other was processed for western blotting. Separated proteins were transferred to a PVDF membrane (Amersham Hybond P 0.2 μm) in a transfer buffer supplemented with 20% methanol. The His$_6$-QueE is detected with an anti-His primary antibody and IRDye-conjugated secondary antibody (LI-COR). The protein bands were visualized with the Odyssey CLx imaging system (LI-COR). To determine relative QueE abundance, band intensities were quantified using ImageJ [67] and normalized to the amount of protein in the uninduced sample.

### Sequence and structural analysis

The protein sequences of QueE orthologs were obtained from Ecocyc [70]. Multiple sequence alignments were performed using Clustal Omega [71,72], and the phylogenetic tree diagrams were generated using Jalview [73]. The consensus sequence was generated using Weblogo [46,74]. The structural model of *Pa*QueE was computed using AlphaFold [73], and the structural alignment of *Bs*QueE, *Ec*QueE, and *Pa*QueE was generated using Pymol [75].

### Supporting information

**S1 Text.  Table A. List of strains used in the study. Table B. List of plasmids used in the study. Table C. List of oligonucleotides used in the study. Fig (i). His$_6$-tagged QueE regulates septation in a dose-dependent manner.** (a) Measurement of cell lengths of *E. coli* MG1655 Δ*lacZYA* (TIM183) cells expressing QueE. Strains containing a plasmid encoding *E. coli* His$_6$-*Ec*QueE (pSY85) or empty vector (pEB52) were grown in supplemented MinA minimal medium for 2 hours (OD$_{600}$ = 0.2–0.3) and induced with IPTG for 3 hours at the indicated concentration. Gray circles represent individual cells, mean cell length values are indicated in red, and the horizontal gray bars represent the median. Data are obtained from four independent experiments using the number of cells indicated by (n) for each inducer concentration.

Statistical analysis was done using t-test displaying significance at $^*P \leq 0.05$, $^{**}P \leq 0.01$, $^{***}P \leq 0.001$, $^{****}P \leq 0.0001$, and "ns" = P >0.05 (b) Representative phase-contrast micrographs of $\Delta lacZYA$ cells expressing QueE at the indicated inducer concentration, scale bar = 5 μm. (c-d) Representative Coomassie gel showing protein levels in total cell lysates at different IPTG concentrations and the corresponding western blot showing His$_6$-QueE levels. e) APB gel-northern blot detecting tRNA$^{Tyr}$ in total RNA samples of $\Delta queE$ (SAM31) cells encoding either WT QueE (pRL03), His$_6$-QueE (pSY85) or an empty vector (pEB52). Cells were grown in supplemented MinA minimal medium for 2 hours and induced with 0.5 mM IPTG for 3 hours. G-tRNA$^{Tyr}$ = unmodified tRNA$^{Tyr}$, Q-tRNA$^{Tyr}$ = Q-modified tRNA$^{Tyr}$. Note that this panel is a montage with the data for pHis$_6$-QueE acquired on another blot, as indicated with a dotted line. (f) Plot showing the relationship between average cell lengths (from panel a) and relative QueE abundance (from panel d), see methods for details. **Fig (ii). Alanine scanning mutagenesis of amino acid residues in *Ec*QueE important for Q biosynthesis.** (a) APB northern blot detecting tRNA$^{Tyr}$ in total RNA samples of $\Delta queE$ (SAM31) cells encoding either WT QueE (pRL03), QueE variants, QueE-R27A (pSY98), QueE-C31A (pSA1), QueE-C35A (pSA2.6), QueE-K66A (pSA10), QueE-Q13A (pSY97), QueE-T40A (pSY99), QueE-K60A (pSA9), QueE-S136A (pSA8), QueE-Q189A (pSY100), QueE-C38A (pSA3) or an empty vector (pEB52). Cells were grown in supplemented MinA minimal medium for 2 hours and induced with 0.5 mM IPTG for 3 hours. G-tRNA$^{Tyr}$ = unmodified tRNA$^{Tyr}$, Q-tRNA$^{Tyr}$ = Q-modified tRNA$^{Tyr}$. (b) Representative phase-contrast micrographs of $\Delta queE$ (SAM31) cells expressing WT QueE and single alanine mutants as indicated above in (a), scale bar = 5 μm. Cells were grown in supplemented MinA minimal medium for 2 hours and induced with 0.5 mM IPTG for 3 hours. (c) Quantification of levels of YFP-tagged QueE and its variants in $\Delta queE$ (SAM96) cells transformed with plasmids encoding WT YFP-QueE (pSY76), YFP-QueE-E15A (pSA73), YFP-QueE-C31A (pSA75), YFP-QueE-C35A (pSA119), or YFP-QueE-K66A (pSA57). Cells were grown in supplemented MinA minimal medium for 5 hours. Data represent the mean and range from two independent experiments, and fluorescence was quantified from the number of cells (n) per sample as indicated. The red dotted line represents the expression level for WT YFP-QueE, Statistical analysis was done using t-test displaying significance at $^*P \leq 0.05$, $^{**}P \leq 0.01$, $^{***}P \leq 0.001$, $^{****}P \leq 0.0001$, and "ns" = P >0.05. (d) APB gel- northern blot detecting tRNA$^{Tyr}$ in total RNA samples of $\Delta queE$ (SAM31) cells encoding either WT QueE (pRL03), YFP-QueE (pSY76) or an empty vector (pEB52). Cells were grown in supplemented MinA minimal medium for 2 hours and induced with 0.5 mM IPTG for 3 hours. Cells expressing YFP-QueE were grown in supplemented MinA minimal medium for 5 hours (please see Methods for details on pSY76). G-tRNA$^{Tyr}$ = unmodified tRNA$^{Tyr}$, Q-tRNA-$^{Tyr}$ = Q-modified tRNA$^{Tyr}$. Note that this panel is a montage with the data for pYFP-QueE acquired on another blot, as indicated with a dotted line. **Fig (iii). Analysis of YFP-tagged variants of *E. coli* QueE and orthologs.** (a) Coomassie stain and (b) western blots of cell lysates of *E. coli* strains encoding YFP-tagged *Ec*QueE (pSY76), its variants C31A (pSA75), C35A (pSA119), $\Delta$E45-W67 (pSA76), and QueE orthologs *Bs*QueE (pSA121) and *Pa*QueE (pSA126). Lysates were prepared from cells grown in supplemented MinA minimal medium for 5 hours (please see Methods for details on pSY76 and derivatives). Cells from a strain carrying a genomic *yfp* reporter (SAM54) are indicated as YFP-only control. **Fig (iv). Q-tRNA formation in cells expressing QueE variants (QueE-C31A and -C35A).** APB gel-northern blot detecting tRNA$^{Tyr}$ in total RNA samples of $\Delta queE$ (SAM31) cells carrying either an empty vector (pEB52), WT QueE (pRL03), QueE-C31A (pSA1) or QueE-C35A (pSA2). G-tRNA$^{Tyr}$ = unmodified tRNA$^{Tyr}$, Q-tRNA$^{Tyr}$ = Q-modified tRNA$^{Tyr}$. Cells were grown in supplemented MinA minimal medium for 2 hours and induced with 0.5 mM IPTG as indicated (+). **Fig (v). Sequence analysis and phylogeny of selected QueE orthologs.** (a) Multiple sequence

alignment of QueE orthologs from *Bacillus subtilis* str 168, *Pseudomonas aeruginosa* PA01, *Klebsiella pneumoniae*, *Escherichia coli K-12 MG1655*, and *Salmonella Typhimurium* str. 14028. The cluster binding motif residues are boxed in black, and residues involved in binding S-adenosyl methionine are highlighted using red circles (single residues) and red boxes (sequential residues). Residues involved in substrate binding (CPH$_4$/6-CP ligation) are denoted by green rectangles (single residues) and green boxes (sequential residues). A single amino acid residue involved in binding the catalytic magnesium ion is indicated by a purple circle. (b) Phylogenetic tree of the QueE orthologs estimated by the neighbor-joining method (BLOSUM62) using Jalview[73]. **Fig (vi). Analysis of dual functions of *Ec*QueE orthologs in their hosts.** (a) Representative phase-contrast micrographs of *E. coli*, *Salmonella Typhimurium*, *Klebsiella pneumoniae*, *Pseudomonas aeruginosa*, and *Bacillus subtilis* expressing their native QueE ortholog. The corresponding plasmid constructs are indicated, p*Ec*QueE (pRL03), p*St*QueE (pSA21), p*Kp*QueE (pSA59), p*Pa*QueE (pSA156), pYFP-*Pa*QueE (pSA154). Empty vectors pEB52 for *E. coli*, *S. Typhimurium*, pBAD33 for *K. pnueumoniae*, pPSV38 for *P. aeruginosa*) were included as controls. *B. subtilis* strains encoding *Bs*QueE (*Bs amyE*::*Bs*QueE, SAA50) and YFP-*Bs*QueE (*Bs amyE*::*yfp-Bs*QueE, SAA52) at the genomic *amyE* locus are indicated and *B. subtilis* strain carrying empty vector pDR111 sequence at *amyE* locus (*Bs amyE*::*Empty*, SAA53) was included as a control. All cultures were grown in supplemented MinA minimal medium and QueE expression was induced using 0.5 mM IPTG in all cases, except for *P. aeruginosa* and *B. subtilis* where IPTG was used at 2 mM and 1 mM, respectively. Scale bar = 5μm. (b) Measurement of cell lengths of *B. subtilis* expressing *Bs*QueE (*Bs amyE*::*Bs*QueE, SAA50), control strain (*Bs amyE*::*Empty*, SAA53), *P. aeruginosa* cells expressing *Pa*QueE (pSA156), or an empty vector (pPSV38). QueE expression was induced using IPTG at 2 mM and 0.5 mM for *P. aeruginosa* and *B. subtilis*, respectively. Gray circles represent individual cells, mean cell length values are indicated in red, and the horizontal gray bars represent the median. Data are obtained from two independent experiments using (n) number of cells per sample as indicated for each strain, Statistical analysis was done using t-test displaying significance at $^*P \leq 0.05$, $^{**}P \leq 0.01$, $^{***}P \leq 0.001$, $^{****}P \leq 0.0001$, and "ns" = P >0.05. (c) FM 4–64 staining of *B. subtilis* cells expressing either *Bs*QueE (*Bs amyE*::*Bs*QueE, SAA50) or control strain (*Bs amyE*::*Empty*, SAA53), QueE expression was induced using 0.5 mM IPTG. Scale bar = 5μm. (d) Quantification of levels of YFP-tagged *Pa*QueE (pSA154) expressed in *P. aeruginosa* with or without induction with 2 mM IPTG, normalized to the background fluorescence in cells carrying an empty vector (pPSV38). Data are obtained from two independent experiments using (n) number of cells per sample as indicated. The corresponding Coomassie-stained gel and western blots show *Pa*QueE protein expression in total cell lysates. (e) Quantification of levels of YFP-tagged *Bs*QueE expressed in *B. subtilis* (Bs *amyE*::*yfp-Bs*QueE, SAA52) with or without induction with 0.5 mM IPTG, normalized to the background fluorescence in control (*Bs amyE*::*Empty*, SAA53). Data are obtained from two independent experiments using (n) number of cells per sample as indicated. The corresponding Coomassie-stained gel and western blots show *Bs*QueE protein expression in total cell lysates. **Fig (vii). Q-tRNA formation in cells expressing QueE deletion mutants (QueE-ΔE45-E51, -ΔV52-W67, and -ΔE45-W67).** APB gel-northern blot detecting tRNA$^{Tyr}$ in total RNA samples of Δ*queE* (SAM31) cells carrying either an empty vector (pEB52), WT QueE (pRL03), QueE- ΔE45-E51 (pSA60), QueE-ΔV52-W67 (pSA61), or QueE-ΔE45-W67 (pSA62). G-tRNA$^{Tyr}$ = unmodified tRNA$^{Tyr}$, Q-tRNA$^{Tyr}$ = Q-modified tRNA$^{Tyr}$. Cells were grown in supplemented MinA minimal medium for 2 hours and induced with 0.5 mM IPTG as indicated (+) for 3 hours. **Fig (viii). Sequence analysis and phylogeny of selected QueE orthologs from gammaproteobacteria** (a) Multiple sequence alignment of 18 QueE orthologs from representative enterobacteria generated using ClustalW. The E45-W67 region is highlighted in red. (b). Phylogenetic tree of

the selected QueE sequences estimated by the neighbor-joining method (BLOSUM62) using Jalview [73]. **Fig (ix). Expression of *Ec*QueE in *P. aeruginosa*.** Representative phase-contrast micrographs of *E. coli* and *P. aeruginosa* cells carrying either p*Ec*QueE (pSA157) or pEmpty (pPSV38 shuttle vector) in the presence or absence of IPTG. IPTG levels of 0.5 mM and 2 mM were used for *E. coli* and *P. aeruginosa*, respectively. Scale bar = 5 μm.
(DOCX)

## Acknowledgments

The authors would like to thank Drs. Mark Goulian, Bryce Nickels, Huizhou Fan, Dieter Schifferli, Jeffrey Boyd, David Dubnau, Simon Dove and Kenn Gerdes for generously sharing strains and plasmids. We are grateful to Drs. David Dubnau and Jeanette Hahn for sharing their expertise and helping us generate IPTG-inducible genomic constructs of *B. subtilis* expressing *B. subtilis* QueE. We thank the past and present members of the Yadavalli and Shah Labs for helpful discussions. We also thank Drs. Premal Shah, Manuela Roggiani, and Mark Goulian for critical reading of the manuscript.

## Author Contributions

**Conceptualization:** Samuel A. Adeleye, Srujana S. Yadavalli.

**Data curation:** Samuel A. Adeleye, Srujana S. Yadavalli.

**Formal analysis:** Samuel A. Adeleye, Srujana S. Yadavalli.

**Funding acquisition:** Srujana S. Yadavalli.

**Investigation:** Samuel A. Adeleye, Srujana S. Yadavalli.

**Methodology:** Samuel A. Adeleye, Srujana S. Yadavalli.

**Project administration:** Samuel A. Adeleye, Srujana S. Yadavalli.

**Resources:** Srujana S. Yadavalli.

**Software:** Samuel A. Adeleye, Srujana S. Yadavalli.

**Supervision:** Srujana S. Yadavalli.

**Validation:** Samuel A. Adeleye, Srujana S. Yadavalli.

**Visualization:** Samuel A. Adeleye, Srujana S. Yadavalli.

**Writing – original draft:** Samuel A. Adeleye, Srujana S. Yadavalli.

**Writing – review & editing:** Samuel A. Adeleye, Srujana S. Yadavalli.

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
