## [Decision Letter · Decision Letter 0]

3 Jan 2024

Dear Dr Yadavalli,

Thank you very much for submitting your Research Article entitled 'Queuosine biosynthetic enzyme, QueE moonlights as a cell division regulator' to PLOS Genetics.

The manuscript was fully evaluated at the editorial level and by independent peer reviewers. The reviewers appreciated the attention to an important topic but identified some concerns that we ask you address in a revised manuscript.

We therefore ask you to modify the manuscript according to the review recommendations. Your revisions should address the specific points made by each reviewer.

Yours sincerely,

Jue D. Wang

Academic Editor

PLOS Genetics

Sean Crosson

Section Editor

PLOS Genetics

Reviewer's Responses to Questions

**Comments to the Authors:**

Reviewer #1: attached

Reviewer #2: This paper from Adeleye and Yadavalli investigates the cell division inhibition activity of Escherichia coli QueE, an enzyme involved in Queuosine (Q) synthesis for tRNA modification. The authors set out to determine whether the roles of QueE in division inhibition and Q synthesis are separable and whether this activity is conserved. Site-directed mutagenesis was performed to generate variants of QueE with amino acid substitutions in the active site region and other areas of the protein, including a region implicated in cell division by bioinformatic analysis. Plasmids expressing the altered proteins were then tested for their ability to promote Q biosynthesis and division inhibition using blotting and microscopy, respectively. The authors also overexpressed QueE orthologs from other bacteria in E. coli and in their native hosts to test for conservation of the division inhibition activity. Close relatives of E. coli produce QueE proteins that block division, whereas more distant relatives (P. aeruginosa and B. subtilis) do not. Overall, the results support a moonlighting function in division inhibition for QueE that is independent of catalytic activity needed for Q synthesis.

The paper is very straightforward, and results clearly presented and described. In general, I agree with the conclusions and that they are supported by the data. However, there are some areas of the paper that would benefit from revision.

Major points:

1) The QueE variants that do not block cell division were tested for stable accumulation based on the fluorescence of YFP fusions. This assay is not sufficient, as the fusion protein could be cleaved with the YFP portion stably accumulating and the QueE portion being degraded. Immunoblots should be performed to ensure that full-length protein for these variants accumulates to levels similar to that of the WT protein.

2) I think it would be worth trying AlphaFold to see if it predicts an interaction between Ec-QueE and Ec-FtsZ and whether the residues in this study implicated in division inhibition lie at the putative interface. If results for this analysis were positive, it would significantly strengthen the paper.

Minor points:

3) Given the number of proteins with moonlighting functions that have now been characterized, I do not think it is necessary to make a major point about challenging the one enzyme, one function concept in the Author Summary.

4) Lines 116-122 are redundant with the Introduction section and could be cut.

5) Line 679: QueE regulates…  QueE inhibits…

6) Figure 2b – increase size of labels.

7) Figure 3 legend. Amino acid residues are “substituted” not “mutated”. Mutation is a term used for genes not proteins.

8) Fig. 3A: Add color to the AAs shown as sticks to make them stand out.

9) Figure 3B: The first column would be better labeled as “Amino acid substitution”, and the entries as “Q13A, E15A, etc….”.

10) Fig. 4A: The cells are way too small to see the Z-rings and QueE localization. An inset with a zoom is needed.

11) Fig 4B: Is QueE in these constructs produced from a gene under different regulation that the IPTG-induced plasmids used in other panels? If so, this should be explicitly indicated.

Reviewer #3: This study by Adeleye and Yadavalli illuminates the contribution of the conserved tRNA modification enzyme QueE to division inhibition in E. coli. Induced in response to cationic antimicrobial peptides (CAP) or magnesium starvation via PhoP/Q, accumulation of the queE gene product leads to filamentation presumably through interaction with components of the divison machinery. QueE colocalizes to the future division site with the conserved bacterial division protein FtsZ. Through a combination of biochemistry, genetics, and comparative genomics, they identify a domain of E. coli QueE that is involved in mediating division inhibition independent of QueE’s role in tRNA modification. Additionally, they provide data suggesting that while QueE overproduction leads to filamentation in E. coli and closely related organisms, it does not modulate division in other bacteria including Pseudomonas and Bacillus subtilis.

While the physiological relevance of having two, apparently disparate functions encoded by a single protein remains unclear, I found experiments and computational analysis indicating that two, separate domains are involved in QueE mediated tRNA modification and division inhibition very compelling. At the same time, the work would benefit from additional quantitative data on queE/QueE expression/accumulation in various conditions including antimicrobial challenge and magnesium starvation. Although the authors provide a (single?) quantitative immunoblot indicating His-QueE fusion protein levels at different IPTG concentrations (Figure S1) and relative fluorescence data from YFP-QueE-WT and various mutant constructs (Figures 4 and S2), how QueE concentrations following CAP treatment or magnesium starvation compare to those in strains encoding IPTG inducible queE constructs is unclear. Moreover, without information on QueE concentration it is impossible to assess whether queE overexpression really has no impact on B. subtilis or P. aeruginosa division, or if levels are too low to impact division in those organisms. The addition of this information would enhance experimental rigor and provide physiological benchmarks with which to assess the results.

Other comments:

1. Line 129: Please clarify whether or not queE is induced in some manner here and other places in the text.

2. Line 136. This section is not particularly clear. It is presented as results but is more generally an explanation of the assays. It would be better to fold into the next section which tests the dose-dependency of QueE on division

3. Line 207: How high do levels need to be to cause filamentation? (i.e. the meaning of ‘high’ is unclear—5 times basal levels? 100x basal levels?)

4. Bacillus subtilis typically grows as a mixture of short and long chains of separated cells. In order to accurately evaluate the impact of BsQueE on division, it is important to use a cell wall stain to resolve septa between individual bacteria.

**Have all data underlying the figures and results presented in the manuscript been provided?**

Reviewer #1: Yes

Reviewer #2: Yes

Reviewer #3: Yes

PLOS authors have the option to publish the peer review history of their article (what does this mean?). If published, this will include your full peer review and any attached files.

Reviewer #1: **Yes: **Jennifer K. Herman

Reviewer #2: No

Reviewer #3: No

---

## [Editor Report · Decision Letter 1]

3 May 2024

Dear Dr Yadavalli,

We are pleased to inform you that your manuscript entitled "Queuosine biosynthetic enzyme, QueE moonlights as a cell division regulator" has been editorially accepted for publication in PLOS Genetics. Congratulations!

Yours sincerely,

Jue D. Wang

Academic Editor

PLOS Genetics

Sean Crosson

Section Editor

PLOS Genetics

Comments from the reviewers (if applicable):

**Data Deposition**

http://datadryad.org/submit?journalID=pgenetics&manu=PGENETICS-D-23-01354R1

**Press Queries**

---

## [Editor Report · Acceptance letter]

13 May 2024

PGENETICS-D-23-01354R1 

Queuosine biosynthetic enzyme, QueE moonlights as a cell division regulator 

Dear Dr Yadavalli, 

We are pleased to inform you that your manuscript entitled "Queuosine biosynthetic enzyme, QueE moonlights as a cell division regulator" has been formally accepted for publication in PLOS Genetics! Your manuscript is now with our production department and you will be notified of the publication date in due course.

With kind regards,

Lilla Horvath

PLOS Genetics

On behalf of:
